# A Study of Small Intestinal Epigenomic Changes Induced by Royal Jelly

**DOI:** 10.3390/cells13171419

**Published:** 2024-08-25

**Authors:** Genki Kobayashi, Takahiro Ichikawa, Takuro Okamura, Tomoyuki Matsuyama, Masahide Hamaguchi, Hideto Okamoto, Nobuaki Okumura, Michiaki Fukui

**Affiliations:** 1Department of Endocrinology and Metabolism, Graduate School of Medical Science, Kyoto Prefectural University of Medicine, Kyoto 602-8566, Japan; genkoba@koto.kpu-m.ac.jp (G.K.); takaichi@koto.kpu-m.ac.jp (T.I.); tomo10@koto.kpu-m.ac.jp (T.M.); mhama@koto.kpu-m.ac.jp (M.H.); michiaki@koto.kpu-m.ac.jp (M.F.); 2Institute for Health Science, R&D Department, Yamada Bee Company, Inc., Okayama 708-0393, Japan; ho1993@yamada-bee.com (H.O.); no1780@yamada-bee.com (N.O.)

**Keywords:** royal jelly, genomics, epigenomics, CUT&Tag, multi-omics

## Abstract

This study explores the impact of royal jelly (RJ) on small intestinal epigenomic changes. RJ, produced by honeybees, is known for its effects on metabolic diseases. The hypothesis is that RJ induces epigenomic modifications in small intestinal epithelial cells, affecting gene expression and contributing to metabolic health. Male *db/m* and *db/db* mice were used to examine RJ’s effects through mRNA sequencing and CUT&Tag methods. This study focused on histone modifications and gene expression changes, with statistical significance set at *p* < 0.05. RJ administration improved insulin sensitivity and lipid metabolism without affecting body weight. GO and KEGG pathway analyses showed significant enrichment in metabolic processes, cellular components, and molecular functions. RJ altered histone modifications, increasing H3K27me3 and decreasing H3K23Ac in genes associated with the G2M checkpoint. These genes, including Smc2, Mcm3, Ccnd1, Rasal2, Mcm6, and Mad2l1, are linked to cancer progression and metabolic regulation. RJ induces beneficial epigenomic changes in small intestinal epithelial cells, improving metabolic health and reducing cancer-associated gene expression. These findings highlight RJ’s potential as a therapeutic agent for metabolic disorders. Further research is needed to fully understand the mechanisms behind these effects and their implications for human health.

## 1. Introduction

Royal jelly (RJ) is a naturally occurring substance synthesized by honeybees and fed to the queen and young worker larvae [1,2]. RJ comprises three major nutrients: proteins, carbohydrates, and lipids, and is a source of various vitamins and minerals. RJ promotes the optimal growth of the queen honeybee and has been reported to exert multiple effects on metabolic diseases, such as glucose intolerance, lipid disorders, and hypertension [3,4,5,6]. Four types of medium-chain fatty acids have been identified in RJ: 10-hydroxy-2-decenoic acid, 10-hydroxydecanoic acid, 2-decenoic diacid, and sebacic acid. These medium-chain fatty acids have been demonstrated to be involved in preventing nonalcoholic fatty liver disease [7]. In that study, RJ administration to *db/db* mice with severe obesity and impaired glucose tolerance due to a lack of leptin receptors was found to increase the number of inflammatory cells in the mucosal lining of the small intestine and alter the expression of genes related to nutrient absorption in the small intestine. Based on the previous study, it is hypothesized that RJ regulates gene expression via epigenomic changes in small intestinal epithelial cells. Nutrient metabolism is finely regulated by transcriptional networks, and disturbances in these networks are closely linked to a variety of diseases, including metabolic diseases. Epigenetic mechanisms such as DNA methylation and histone modifications play an important role in the regulation of these transcripts and are involved in disease-related transcriptional signatures, which are also involved in determining signatures. Recent studies are revealing the relationship between genetic and epigenetic markers associated with obesity [8]. For instance, it has been shown that n-3 PUFA (polyunsaturated fatty acids) supplementation is associated with altered methylation patterns at specific gene loci in subjects on energy-restricted diets [9]. The impact of epigenetics on lipid metabolism has also been extensively studied. One study demonstrated that the Mediterranean diet altered DNA methylation patterns at specific genes and improved the expression of genes associated with diabetes and inflammatory pathways [9]. The aforementioned reports and previous studies suggest that changes in nutritional intake may promote epigenomic alterations in the small intestine, potentially contributing to various metabolic disorders. The Cleavage Under Targets and Tagmentation (CUT&Tag) method, an enzyme-linked strategy that provides an efficient high-resolution sequencing library for profiling diverse chromatin components, is employed [10]. In the CUT&Tag method, chromatin proteins are bound in situ by specific antibodies, followed by the binding of a protein A-Tn5 transposase fusion protein. The utility of the CUT&Tag method has been demonstrated by profiling histone modifications, RNA polymerase II, and transcription factors in low cell numbers and single cells. In this study, RJ-induced epigenomic changes in small intestinal epithelial cells were tested using small intestinal mRNA sequencing and the CUT&Tag method to validate the hypothesis.

## 2. Materials and Methods

### 2.1. Murine Models

The entirety of the experimental methodologies received approval from the Committee for Animal Research at Kyoto Prefectural University of Medicine, Japan (approval designation: M2024-66). This study was characterized by randomization and investigator blinding.

Seven-week-old male C57BLKS/JIar-+*Lepr^db^*/+*Lepr^db^* (*db/db*) (Jax Stock Number 000662) and C57BLKS/JIar-*m*+/+*Lepr^db^* (*db/m*) mice were purchased from Shimizu Experimental Materials in Kyoto, Japan, and reared in a strictly controlled, pathogen-free environment. The RJ powder used in this study was derived from RJ treated with alkaline proteases to eliminate allergens [10]. It was standardized to include a minimum of 3.5% trans-10-hydroxy-2-decenoic acid and a minimum of 0.6% 10-hydroxydecanoic acid (Lot: TRP-M-210728-1; Yamada Bee Company, Inc., Okayama, Japan). Four 8-week-old murine models were allocated to the following groupings: (1) *db/m* mice, (2) *db/db/* mice, and (3) *db/db*+RJ mice (5% weight) [11]. We have previously tested three different concentrations of royal jelly (0.2%, 1%, and 5% weight) for NAFLD prevention. Since 5% was the most effective in preventing NAFLD, only 5% was used in this study. To determine the sample size needed to compare the groups, a power analysis was performed using the mean and standard deviation of the insulin tolerance test (ITT) for *db/db* and *db/db*+RJ mice in a prior experiment. Based on a significance level of 0.05 and a power of 80%, it was determined that a minimum of four subjects per group were required to detect statistically significant differences. Based on the results, the experiment was to be conducted using four mice per group. Mice were fed a standard nutritional regimen (ND; 345 kcal/100 g, fat accounting for 4.6% of kcal; obtained from CLEA, Tokyo, Japan) for 8 weeks, starting at 8 weeks of age. Pair feeding was performed to ensure that there was no difference in the amount of *db/db/*mice and *db/db*+RJ mice fed. Subsequently, the subjects underwent an overnight fasting period and were euthanized at sixteen weeks of age via exposure to an anesthetic amalgamation (comprising 4.0 mg/kg midazolam, 0.3 mg/kg medetomidine, and 5.0 mg/kg butorphanol) (Figure 1A).

### 2.2. Quantification of Voluntary Wheel Running

Murine subjects were singularly accommodated within enclosures, each equipped with a running wheel (Model MK-713; Muromachi Kikai, Tokyo, Japan). The cumulative rotations of the incorporated running wheel were meticulously logged throughout each nocturnal 12 h phase, utilizing specialized software (CompACT AMS Ver.3: Muromachi Kikai) interfaced with the computational apparatus connected to the respective running wheels (n = 4). The assessment of locomotor activity was diligently performed for a quintet of days preceding the scheduled euthanasia.

### 2.3. Analytical Methodologies and Tolerance Tests for Glucose and Insulin

Murine models, aged fifteen weeks, were subjected to ITT (dispensed at 0.5 U/kg of body weight) following a 5-h fast (n = 4). Venous blood specimens were procured from the caudal vein. Glycemic indices were ascertained employing a glucometer (Model Gultest mint II; Sanwa Kagaku Kenkyusho, Nagoya, Japan). Glycemic indices were meticulously monitored at intervals of 0, 15, 30, 60, and 120 min post-injection. The area under the curve (AUC) of the yielded ITT data was subjected to analytical scrutiny.

### 2.4. Serum Biochemical Analysis

Blood specimens were procured from fasted murine models via cardiac perforation during the process of euthanasia, and the serum specimens were isolated after centrifugation conducted at 14,000 revolutions per minute for 10 min at 4 °C. The isolated serum was preserved at −30 °C pending dispatch to an analytical subcontractor. Concentrations of triglycerides (TG) and total cholesterol were quantified utilizing enzymatic methodologies (TG, GK-GPO) [12]. Biochemical evaluations (n = 4) were executed at the FUJIFILM Wako Pure Chemical Corporation (Osaka, Japan).

### 2.5. Histopathological Evaluation of Jejunum and Colon

Jejunum and colon samples, meticulously excised from murine models, were promptly conserved in 10% buffered formaldehyde and Carnoy’s solution for 24 h at 22 °C. Following this, they were embedded in paraffin, partitioned into 4 µm thick sections, and underwent staining procedures with hematoxylin and eosin (HE) and periodic acid-Schiff (PAS) stain. Visualization of the stained sections was achieved using a fluorescence microscope (BZ-X710; Keyence, Osaka, Japan). The height and width of the villi and the crypt depth were ascertained using HE-stained sections at five distinct locations per slide for each aggregation of 10 specimens, leveraging ImageJ software (Version 1.53 k, NIH, Bethesda, MD, USA). Mucin granules and goblet cells (PAS+) were quantified and represented as the average count of goblet cells (PAS+) per 10 crypts using ImageJ software, as formerly specified (n = 4) [13].

### 2.6. Analysis of mRNA Sequencing of the Jejunum

The jejunum from murine models that underwent a 16-h fasting period were excised and instantaneously preserved in liquid nitrogen. The specimens were homogenized in ice-cold QIAzol Lysis Reagent (Qiagen, Venlo, The Netherlands) at 4000 revolutions per minute for 2 min within a ball mill, followed by the extraction of total RNA following the manufacturer’s prescribed protocol. A complementary DNA (cDNA) library was synthesized utilizing the TruSeq^®^ Stranded mRNA kit (Qiagen, Carlsbad, CA, USA). Paired-end sequencing was executed on the Illumina NovaSeq6000 platform (n = 4).

Differentially expressed genes (DEGs) were further annotated by Gene Ontology (GO) functional enrichment [10] and Kyoto Encyclopedia of Genes and Genomes (KEGG) pathway analysis [14]; these analyses facilitate the understanding of the biological functions of genes and the utility of the biological system within large-scale molecular datasets. The number of genes is the number of genes enriched in GO terms and KEGG pathways. Gene ratio is the percentage of total DEGs in a given GO term or KEGG pathway. Results were sorted according to the proportion of enriched genes in each entry, and the top 10 results were visualized as dot plots. Significance levels were adjusted using the Benjamini–Hochberg method.

To scrutinize the functional enrichment of prolific genes within cells and nuclei, each gene was ranked based on log fold change, and gene set enrichment analyses (GSEA) were conducted on gene markers derived from prior studies [15]. Within these analyses, a positive normalized enrichment score (NES) signifies gene enrichment, whereas a negative NES denotes gene depletion.

### 2.7. Quantitative Real Time-PCR

The extraction of total RNA from the jejunum of murine models was converted to cDNA, and gene expression was analyzed with quantitative real-time PCR (qRT-PCR) (n = 4). The PCR primers used in this study, *Smc2*, *Mcm3*, *Ccnd1*, *Rasal2*, *Mcm6*, and *Mad2l1I*, are listed in Appendix A. The expression of each gene was normalized with *Gapdh* expression. The mRNA expression was measured with TaqMan Fast Advanced Master Mix (ThermoFisher Scientific, Waltham, MA, USA) according to the manufacturer’s protocol. The reactions were carried out in a StepOnePlus real-time PCR system (Applied Biosystems, Foster City, CA, USA) with the following thermal cycling conditions: 50 °C for 2 min, 95 °C for 20 s, followed by 40 cycles of 95 °C for 1 s, and 60 °C for 20 s. The cycle threshold (CT) values were determined using StepOnePlus Software v2.3 (Applied Biosystems). Data were analyzed using the 2^−ΔΔct^ method, and *Smc2*, *Mcm3*, *Ccnd1*, *Rasal2*, *Mcm6*, and *Mad2l1I* mRNA levels were calculated by comparing with *db/db* mice, each normalized to *Gapdh* mRNA as endogenous controls.

### 2.8. Mass Spectrometry for Histone Modifications (Mod-Spec)

Small intestinal epithelial cells were isolated using previously published protocols [16]. Histone acetylation and methylation analysis were conducted using the Mod Spec method, which includes histone extraction, protease digestion, and mass spectrometry. This involved a triple quadrupole mass spectrometer (TSQ Quantiva; Thermo Scientific, MA, USA) directly coupled with a Dionex UltiMate 3000 nano-liquid chromatography system. The analysis was performed by Active Motif (CA, USA) (n = 4) [11].

### 2.9. NON-TiE-UP Cleavage Under Targets and Tagmentation

The foundational kit used for CUT&Tag was a CUT&Tag-IT Assay Kit (n = 1) (Active Motif). After rinsing the zona-free blastocysts with phosphate-buffered saline combined with 0.01% (*w*/*v*) polyvinyl alcohol and 1% (*v*/*v*) Protease Inhibitor Cocktail (PIC), they were individually placed into Antibody Buffer, enriched with the target primary antibody, digitonin, and PIC, situated in a round bottom-shaped 96-well plate. An overnight incubation at 4 °C with gentle agitation was employed, with a negative control being established by omitting the primary antibodies. Following primary antibody interaction, blastocysts were moved to wells containing Dig-Wash buffer, a secondary antibody, digitonin, and PIC and were incubated at room temperature. After washes, the blastocysts were relocated to wells containing Dig-300 Buffer, pA-Tn5 transposomes, digitonin, and PIC and once again incubated at room temperature. After washes, blastocysts were moved to individual microcentrifuge tubes containing Tagmentation Buffer with digitonin and PIC and incubated at 37 °C. After tagmentation, EDTA, SDS, and proteinase K were added and underwent a series of incubation and heating steps. SPRIselect beads were incorporated, vortexed, and incubated, followed by magnetic separation and subsequent washes with ethanol. After drying, a DNA Purification Elution Buffer was added, and the liquid containing tagmented DNA was collected. PCR amplification of sequencing libraries was conducted using tagmented DNA and indexing primers according to the manufacturer’s instructions. The PCR libraries experienced a series of temperature conditions, followed by post-PCR library purification with SPRIselect beads and ethanol washes. The sequencing libraries were ultimately eluted in a DNA Purification Elution Buffer. The paired-end 38 bp sequence reads (PE38) obtained by Illumina Sequencing was aligned to the genome using the default settings of the BWA algorithm, and alignment information was stored in binary alignment map (BAM) format. Only reads satisfying specific criteria were considered for subsequent analyses, and duplicate reads were removed. The MACS3 peak calling algorithm identified genomic regions with high transposition/tagging events, and fragment density was assessed by dividing the genome into specified bins and determining the number of fragments within [17].

### 2.10. Statistical Examination

One-way analysis of variance and Holm–Šídák’s multiple comparisons test were employed to compare the results among different groups. An unpaired *t*-test was utilized to compare the results between the two groups. Statistical significance was determined at *p* < 0.05. Figures were generated using GraphPad Prism version 9.0 software (San Diego, CA, USA).

## 3. Results

### 3.1. Effect of Royal Jelly on Body Weight and Food Intake

To determine the impact of RJ on body weight and food intake, experiments were conducted using *db/m* mice, *db/db* mice, and *db/db* mice fed a diet containing RJ (*db/db*+RJ mice) (Figure 1A). The administration of RJ did not change the body weight, and no significant differences in oral intake were observed with pair feeding (Figure 1B).

### 3.2. Royal Jelly Improves Locomotor Activity

To assess the effect of RJ on locomotor activity, locomotor activity measured by the voluntary wheel running was less in *db/db*+RJ mice than in *db/m* mice but higher than in *db/db* mice, in both light and dark periods (light period: *db/m*, 409 ± 185/12 h, *db/db*, 3.3 ± 3.1/12 h, *db/db*+RJ, 114 ± 26/12 h; dark period: *db/m*, 3350 ± 687/12 h, *db/db*, 32 ± 3.1/12 h, *db/db*+RJ, 195 ± 45/12 h) (Figure 1C).

### 3.3. Royal Jelly Enhances Insulin Sensitivity and Reduces Cholesterol and Triglyceride Levels

To evaluate the effect of RJ on insulin sensitivity, ITT was conducted on the different groups. In ITT, RJ administration significantly improved insulin sensitivity (Figure 1D). Moreover, to investigate the impact of RJ on lipid profiles, serum total cholesterol and triglyceride levels were measured. Serum total cholesterol and triglyceride levels were significantly lower in *db/db*+RJ mice than in *db/db* mice, and TG levels were as low as in *db/m* mice (Figure 1E).

### 3.4. GO and KEGG Pathway Enrichment Analysis of Small Intestine mRNA

Next, mRNA-seq of the small intestine was performed. Sequencing coverage statistics are listed in Table 1.

To reveal potential gene functions and enrichment pathways in *db/db* and *db/db*+RJ mice, GO term and KEGG pathway enrichment analyses were performed. In the GO enrichment analysis, the top 10 terms of biological processes, cellular components, and molecular functions were visualized in dot plots. In the KEGG enrichment analysis, the top 20 pathways were visualized as dot plots, respectively (Figure 2A). The redder the color of the dots, the lower the *p*-value, indicating a higher enrichment of GO terms and KEGG pathways. The results indicated that metabolic processes, cellular components, and molecular function were closely related to the improvement of metabolic disorders in RJ. In addition, more significant enrichment was found in metabolic processes in terms of biological processes, in binding in terms of cellular processes, and in organelles in terms of molecular function. The results of the KEGG analysis showed that 199 statistically significant relative pathways were shown. In the top 20 significantly enriched potential pathways with the highest gene proportions, the “retinol metabolism,” “PI3K-Akt signaling pathway,” “drug metabolism,” “cell cycle,” and “AMPK signaling pathway” pathways were enriched. The results of this study showed that RJ might be involved in the improvement of metabolic disorders through pathways.

### 3.5. The Expression of the Gene Set Associated with the G2M Checkpoint Was Reduced by Royal Jelly

Transcription data sets were subjected to Gene Set Enrichment Analysis (GSEA) to determine biological significance. Among the significantly enriched gene sets, those with *p*-values less than 0.05 were selected for significance. *db/db* mice were enriched for fatty acid metabolism, adipogenesis, bile acid metabolism, and other gene sets compared to db/m mice enriched in *db/db* mice, whereas no enriched gene sets were detected in db/m mice (Appendix A). Next, compared to *db/db*+RJ mice, gene sets related to the G2M checkpoint and coagulation were enriched in *db/db* mice (Appendix A). In addition, *db/db*+RJ mice had enriched gene sets associated with the p53 pathway and reactive oxygen species pathway (Appendix A). Since the gene sets involved in the G2M checkpoint were significantly enriched in *db/db* mice compared to db/m and *db/db*+RJ mice, we investigated further (Figure 2B). First, the expression of a group of genes involved in the G2M checkpoint was shown in a volcano plot (Figure 2C). Compared to *db/db* mice, the expression of *Smc2*, *Mcm3*, *Ccnd1*, *Rasal2*, *Mcm6*, and *Mad2l1* genes was significantly lower in *db/db* + RJ mice. Their gene expression analyzed by qRT-PCR in *db/db*+RJ mice was lower than that in *db/db* mice, as well as mRNA-seq with statistical significance (Figure 2D). The significant differences in cell cycle signaling pathways in the KEGG pathway between *db/db* and *db/db*+RJ mice, as well as differences in gene groups involved in the G2M checkpoint in GSEA, suggest that RJ may alter the expression of these gene groups. We hypothesized that this reduction in G2M checkpoint gene expression is related to the enrichment of cell cycle pathways and epigenetic regulation by RJ.

### 3.6. Relative Abundance of Histone Modifications in Small Intestinal Epithelial Cells

To determine whether RJ treatment causes changes in the chromatin structure of small intestinal epithelial cells, changes in the relative amounts of posttranslational modifications were examined between *db/db* and *db/db*+RJ mice by Mod Spec (Figure 3A). Histones corresponding to antibodies that have been validated to perform well in the CUT&Tag assay include H3K4me2, H3K4me3, H3K9ac, H3K27me3, and H3K36me3. Because the differences between *db/db* and *db/db*+RJ mice were greatest for H3K23ac among acetylation and for H3K27me3 among methylation, further validation of those two histone modifications was performed (Figure 3B).

### 3.7. Royal Jelly Alters Distribution and Increases H3K27me3 and H3K9Ac Peak Tags

To investigate the impact of RJ on the distribution and frequency of histone modifications H3K27me3 and H3K9Ac, we analyzed peak numbers across different genomic regions using pie charts (Figure 4A). Sequencing coverage statistics are listed in Table 2. The genomic distribution showed the highest number of peaks in the distal intergenic region for H3K27me3 and in the intron region for H3K23ac. In the H3K23Ac analysis, *db/db*+RJ mice exhibited a significantly higher number of peak tags compared to db/m and *db/db* mice. Conversely, *db/db* mice had significantly more peak tags in H3K27me3 than db/m and *db/db*+RJ mice (Figure 4B).

### 3.8. Royal Jelly Enhances Correlation and Modifies Peak Sizes of Histone Modifications

To further explore histone modification changes, we examined the correlation between peak tags using heatmaps and scatterplots showing Pearson correlation coefficients (Figure 4C,D). Strong correlations were observed in H3K27me3 among the three groups. Additionally, we compared peak sizes by analyzing tag distributions across gene bodies (with 2 kb flanking regions), merged peak regions (all peak regions; ±5 kb), and transcription start sites (TSS; ±5 kb). Mean plots and heatmaps of six groups indicated that peak sizes were higher in *db/m*, followed by *db/db* and *db/db*+RJ mice in gene bodies, merged peak regions, and promoters (Figure 5A–C).

### 3.9. Quantitative Comparison of Epigenetic Modifications in Key Genes

Finally, to quantitatively assess the level of epigenetic modifications in specific genomic regions, we calculated the integrated signal of histone modifications in the *Smc2*, *Mcm3*, *Ccnd1*, *Rasal2*, *Mcm6*, and *Mad2l1* genes, which were downregulated in *db/db*+RJ mice compared to *db/db* mice by mRNA-seq. The integrated signal of H3K27me3 was higher in *db/db*+RJ mice, whereas the integrated signal of H3K23Ac was lower (Figure 6A,B).

## 4. Discussion

In this study, as in previous research, RJ was found to improve glucose intolerance and dyslipidemia. Furthermore, G2M checkpoint-related gene expression in the small intestine was altered, which was correlated to histone modifications induced by RJ.

Using the CUT&Tag method to analyze histone methylation and acetylation in small intestinal epithelial cells, the number of peak regions for histone H3 lysine 27 trimethylation (H3K27me3), which is involved in transcriptional repression, was low in *db/db* mice, whereas the number of peak regions for histone H3 lysine 23 acetylation (H3K23Ac), which is responsible for transcriptional activity, was high [12,13]. On the other hand, the RJ administration caused those responses to go in the opposite direction. This finding suggests that RJ acts on small intestinal epithelial cells and causes changes to histone modifications.

In epigenomic analysis, open chromatin structures are detected using techniques such as ChIP-seq, FAIRE-seq, and ATAC-seq, which combine chromatin immunoprecipitation (ChIP) and next-generation sequencing to examine histone modifications and transcription factor and cofactor binding. Genomic regions that regulate gene transcription, such as enhancers and promoters, are known to exhibit histone acetylation or methylation and open chromatin structures that lack histone proteins. The CUT&Tag method used in this study enables analysis comparable to conventional ChIP sequencing with a smaller number of cells and represents a major advance in the analysis of epigenomic alterations [10]. Lysine residues in the N-terminal tail of histones can undergo acetylation, a process that relaxes chromatin structure, thereby enabling transcription factors to access and facilitate gene transcription. As a result, histone acetylation is commonly linked to active gene expression. Abnormal patterns of histone acetylation and deacetylation have been associated with various pathological conditions, including inflammatory and degenerative diseases [14]. The relationship between the pathogenesis of diabetes and histone modifications has been pointed out in many studies, with particular attention to histone modifications in the pancreatic islets and intestine, although there have been various reports in the fields of cancer and inflammatory bowel disease [15,16]. However, this study focused on RJ and small intestinal epithelial cells and, for the first time, revealed changes associated with RJ in the regulation of G2M checkpoint-related gene expression. The DNA damage checkpoint in the G2/M phase provides an opportunity to repair DNA damage before the cell proceeds to mitosis. The activity of the Cyclin B-cdc2 complex is critical for G2/M phase regulation and is inactivated by Wee1 and Myt1. DNA damage activates DNA-PK/ATM/ATR kinases p53, regulates the cell cycle through various genes, and prevents cancer formation. p53-dependent WIP1 phosphatase mutations are associated with cancer development, and mutations in DNA repair proteins have also been implicated in tumor suppressor functions. *Smc2*, *Mcm3*, *Ccnd1*, *Rasal2*, *Mcm6*, and *Mad2l1* were among the genes whose gene expression in the small intestine was significantly altered by RJ administration. Smc2 is a crucial component of the condensin complex, playing a vital role in chromatin packaging, processing DNA damage before cell division, ensuring proper chromosome segregation, and maintaining chromosome stability [17]. SMC2 has a dual role in cancer development, being involved in mitotic cell division and potentially promoting cancer. For instance, SMC2 gene knockdown was shown to suppress tumor growth in colorectal cancer [18], and its mRNA expression levels are notably higher in human pancreatic cancer tissue compared to non-tumor tissue [19]. Cyclin D1 (CCND1) is a key regulator of cell proliferation, with its overexpression linked to the development and progression of several malignancies [20]. Additionally, CCND1 overexpression is a significant mechanism of therapeutic resistance in various cancers [21]. The MCM complex (minichromosome maintenance protein complex) is an essential DNA helicase for genomic DNA replication, consisting of six subunits (Mcm2 to Mcm7) forming a heterohexamer [22,23]. MCM is critical for cell division and is targeted by various checkpoint pathways. Mcm3 is overexpressed in most tumors in human studies [24], and MCM6 expression is positively correlated with cell proliferation, migration, invasion, and immune response in numerous cancer types, including breast cancer [25], hepatocellular carcinoma [26], lung cancer [27], and esophageal squamous cell carcinoma [28]. RASAL2, a member of the RAS GTPase-activating protein family, negatively regulates RAS by catalyzing the hydrolysis of RAS-GTP to RAS-GDP. Depending on the cellular context and type of stimulation, RASAL2 can have pro- or anti-tumorigenic effects and is involved in tumor progression in colorectal cancer [29]. MAD2L1 (mitotic arrest defect 2 like 1) is located on human chromosome 4q27 and is part of the MAD family, which comprises components of the mitotic spindle assembly checkpoint. MAD2L1 expression is significantly higher in colon cancer tissue and cell lines compared to adjacent epithelial tissue and normal intestinal epithelial cell lines [30]. The genes whose expression was reduced by RJ administration are all related to cancer onset and progression. Moreover, RJ administration increased the H3K27me3 peak and decreased the H3K23Ac peak in these genes, indicating that RJ may alter gene expression through epigenomic changes.

Few studies have explored the expression of G2M checkpoint-related genes in the small intestine, but recent findings suggest that irradiation of small intestinal epithelial cells increases the expression of both G2M checkpoint-related and DNA repair-related genes [31]. The previous research showed that overeating and inactivity in *db/db* mice cause intestinal microbiota disturbances, or dysbiosis, leading to inflammation in the small intestine’s mucosal lining [7]. This inflammation might contribute to the increased expression of G2M checkpoint-related genes. One of the limitations of this study is that the effects of RJ were examined only in the small intestine. Further studies in organs other than the small intestine would enhance the anti-inflammatory effects of RJ. Further research is needed to understand how the reduced expression of cancer-related genes associated with RJ administration improves glucose intolerance and lipid abnormalities. Furthermore, RNA-seq analysis was performed n = 4 to identify important gene expression changes, but the CUT&Tag assay performed to complement this was only n = 1 per group. The CUT&Tag results are preliminary and intended to provide an initial assessment of epigenetic modifications. CUT&Tag results are preliminary and are intended to provide an initial assessment of epigenetic modifications. Therefore, the CUT&Tag data lack technical replicates and are insufficient for statistical validation. In future studies, the CUT&Tag assay should be performed on a larger number of samples to improve the reproducibility and reliability of epigenetic changes. It is also important to use other complementary methods (e.g., ChIP-Seq, ATAC-Seq) in combination to analyze the observed epigenetic changes in more detail and to confirm the consistency of the results.

## 5. Conclusions

In summary, this study is the first to demonstrate that changes in small intestinal gene expression due to RJ administration are mediated by epigenomic changes. Additionally, RJ administration reduced the expression of genes associated with cancer initiation and progression, providing valuable insights into the health benefits of RJ administration.

## Figures and Tables

**Figure 1 cells-13-01419-f001:**
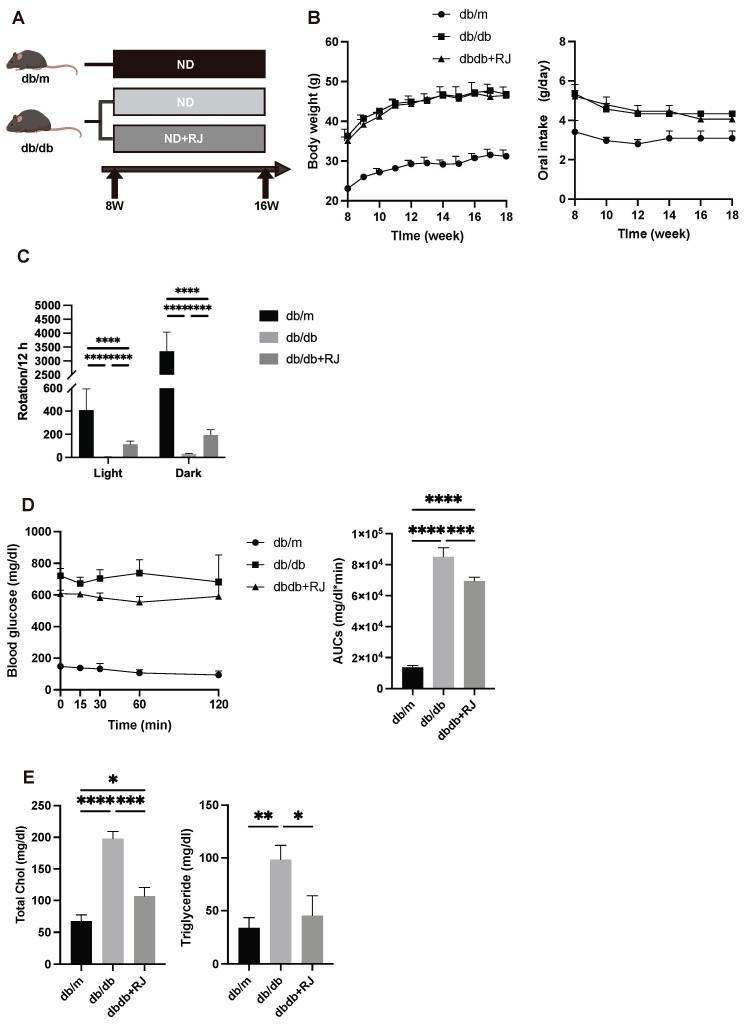
**The administration of royal jelly improved glucose and lipid impairment.** (**A**) Experiment schedule. *db/db* mice were randomly divided into the *db/db* group and the *db/db*+RJ group. (**B**) Changes in body weight and daily oral intake (n = 4). (**C**) The numbers of rotation in the light and dark phases using the running wheel (n = 4). (**D**) Results of the insulin tolerance test (0.5 U/kg body weight) for 15-week-old mice and the AUC analysis (n = 4). (**E**) Serum concentration of total cholesterol and triglycerides (n = 4). Data are represented as the mean ± SD values. Data were analyzed using Welch’s *t*-tests. * *p* < 0.05, ** *p* < 0.01, *** *p* < 0.001, and **** *p* < 0.0001.

**Figure 2 cells-13-01419-f002:**
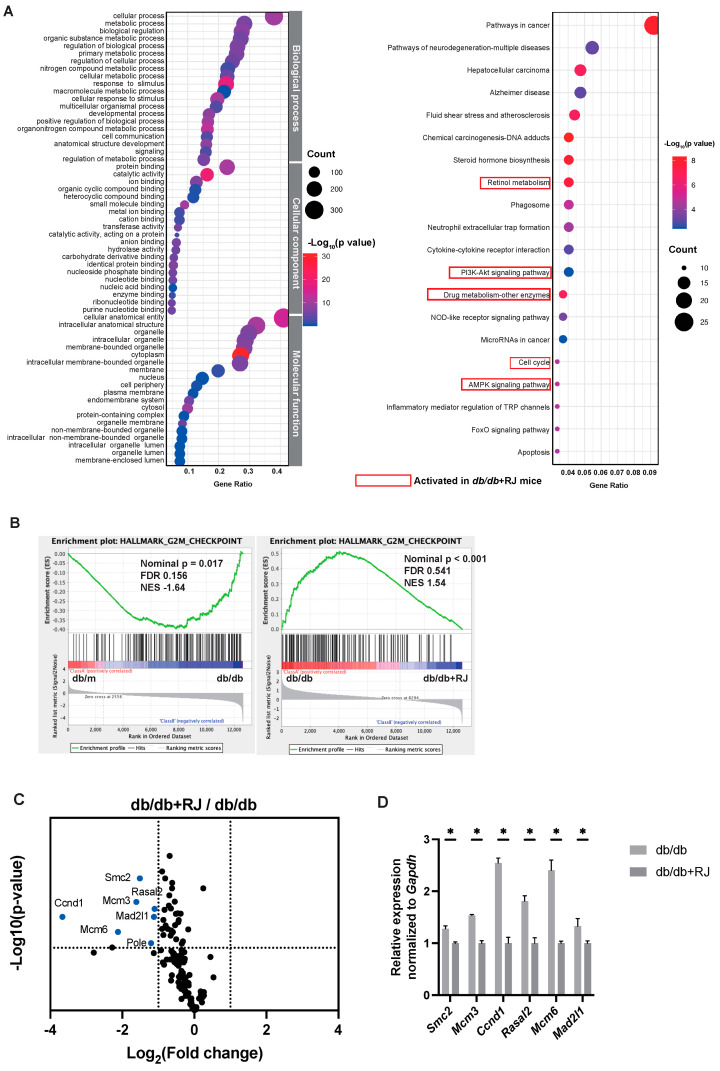
**Gene expression in the small intestine.** (**A**) Dot plots of GO enrichment analysis include the top 10 significant enrichment terms of three domains (ranked by gene ratio): biological process, cellular component, and molecular function. Dot plot of KEGG pathways (ranked by gene ratio). The dot size represents the number of genes belonging to each pathway. The color gradient is related to the level of significance, adjusted with the Benjamini–Hochberg method. The *x*-axis represents the gene proportion enriched in each entry, and the *y*-axis shows the enrichment degree according to the adjusted *p*-value. (**B**) The top of the figure plots the enrichment score for each gene, while the bottom of the plot shows the value of the ranking metric moving down the list of ranked genes in the small intestine. *y*-axis: value of ranking metric; *x*-axis: rank of all genes. (**C**) Volcano plot showing the magnitude and significance of differences in gene expression levels related to the G2M checkpoint of the small intestine in *db/db* mice and *db/db*+RJ mice. blue, fc > 0.5, and raw *p*-value < 0.05. (**D**) The expression of *Smc2*, *Mcm3*, *Ccnd1*, *Rasal2*, *Mcm6*, and *Mad2l1* normalized to *Gapdh* was analyzed by qRT-PCR. Data are represented as the mean ± SD values. Data were analyzed using Welch’s *t*-test. * *p* < 0.001.

**Figure 3 cells-13-01419-f003:**
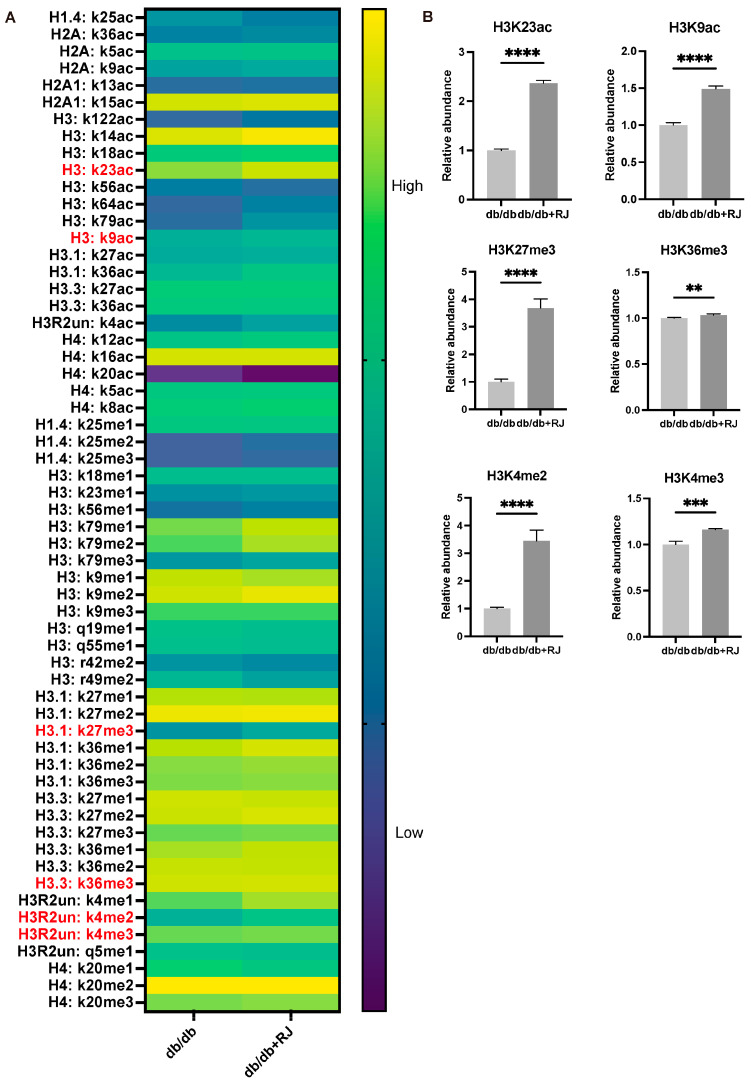
**Histone modifications in small intestinal epithelial cells isolated from *db/db* and *db/db*+RJ mice were determined by Mod Spec.** (**A**) Heat map of the detected histone modifications. Histones in red have been validated by the CUT&Tag assay. (**B**) Relative abundance of histone modification. The average of the present ratio of *db/db* mice was set to 1 (n = 4). Data are represented as the mean ± SD values. Data were analyzed using Welch’s *t*-tests. ** *p* < 0.01, *** *p* < 0.001, and **** *p* < 0.0001.

**Figure 4 cells-13-01419-f004:**
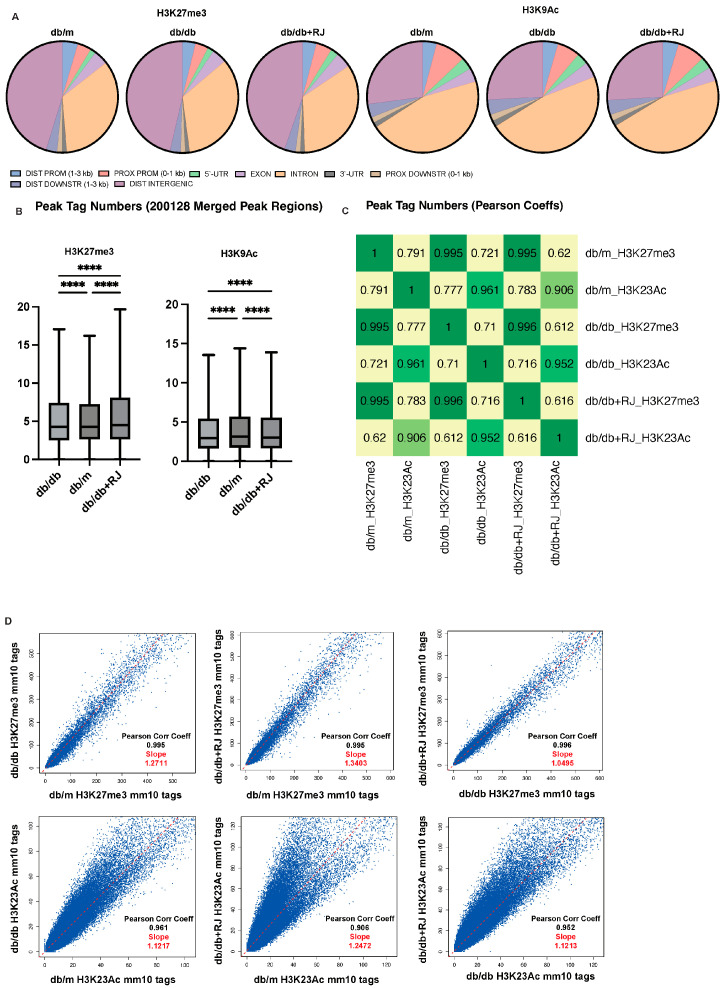
**Global histone modifications via gene bodies in small intestinal epithelial cells.** (**A**) Global occupancy of histone modifications H3K27me and H3K23Ac in different regions of the genome, and the numbers were the number of peaks for each histone mark. (**B**) Tag distributions across target regions such as gene bodies (2 kb upstream and downstream of the transcription start site and transcription end site, respectively) are presented both as average plots (average of values for all target regions in the *y*-axis) and as heatmaps (values in the *z*-axis as color, regions in the *y*-axis). For the heatmaps, the data are clustered using the k-means algorithm (5 clusters indicated by C1–C5) and sorted by decreasing average values inside each cluster. (**C**) Tag distributions across target regions, such as Merged Regions (=all peak regions; ±5 kb), are presented both as average plots (average of values for all target regions in the *y*-axis) and as heatmaps (values in the *z*-axis as color, regions in the *y*-axis). For the heatmaps, the data are clustered using the k-means algorithm (5 clusters indicated by C1-C5) and sorted by decreasing average values inside each cluster. (**D**) Tag distributions across target regions such as TSS (±5 kb) are presented both as average plots (average of values for all target regions in the *y*-axis) and as heatmaps (values in the *z*-axis as color, regions in the *y*-axis). For the heatmaps, the data are clustered using the k-means algorithm (5 clusters indicated by C1–C5) and sorted by decreasing average values inside each cluster. The overlap between peaks was displayed in the VENN diagram. Data are represented as the mean ± SD values. Data were analyzed using Welch’s *t*-test. **** *p* < 0.0001.

**Figure 5 cells-13-01419-f005:**
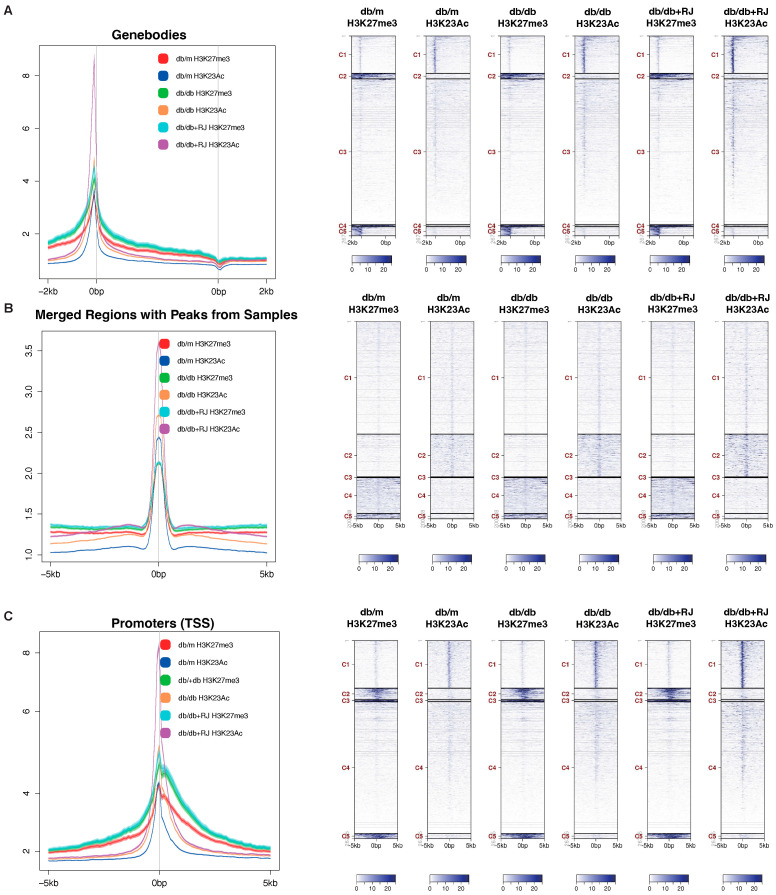
**Multi-dimensional Analysis of Sample Correlations and Distributions in Tagged Regions.** (**A**) This heatmap showed the Pearson correlation coefficients of all pairwise comparisons, colored from dark green (high correlation) to yellow and red (low or no correlation). (**B**) For each pairwise comparison, a scatter plot was generated, plotting the tag numbers of two samples for each merged region. In addition, the slope is a measure of the average ratio of tag numbers between the two samples. (**C**) The boxed area represents the center two quartiles (notched line = median), while the whiskers show the top and bottom quartiles without outliers.

**Figure 6 cells-13-01419-f006:**
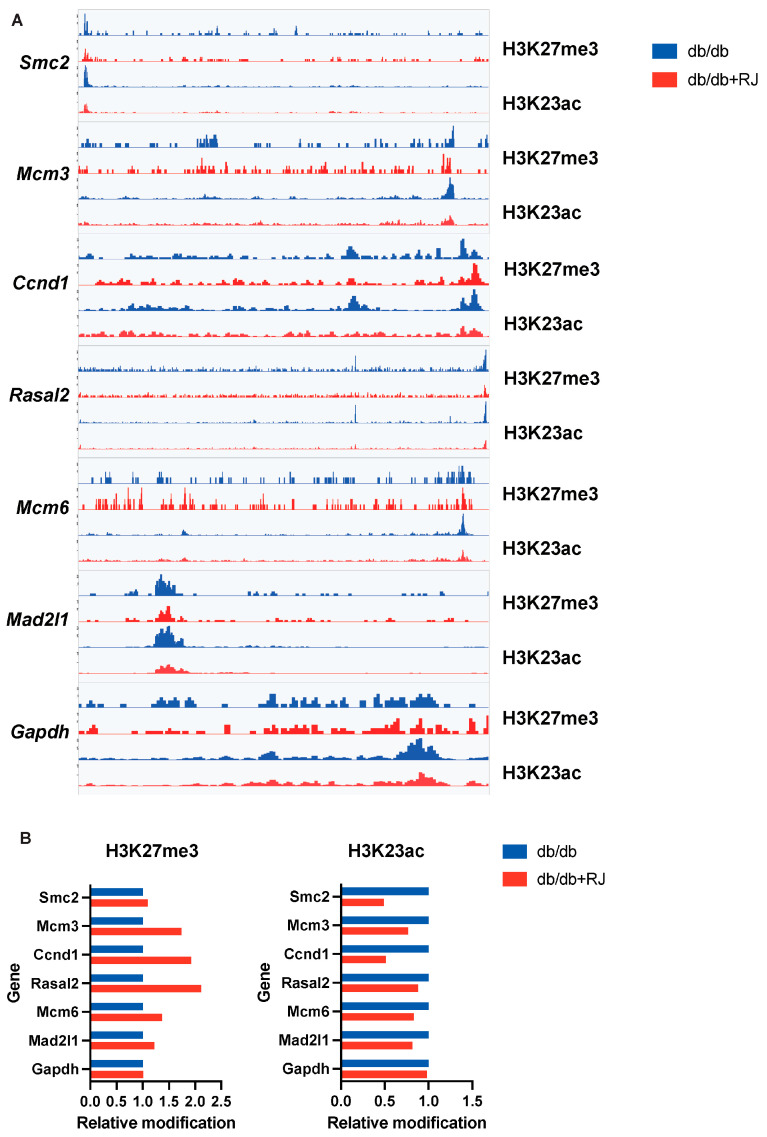
**Characteristics of accessible chromatin in small intestinal epithelial cells of *db/db* and *db/db*+RJ Mice.** (**A**) Genome browser screenshot showing *db/db* (blue) and *db/db*+RJ mice (red). The upper row is H3K27me3, and the lower row is H3K23Ac. CUT&Tag signal from experiments in small intestinal epithelial cells; *Smc2*, *Mcm3*, *Ccnd1*, *Rasal2*, *Mcm6*, *Mad2l1* (six genes with statistically significant differences in expression among genes associated with the G2M checkpoint); and *Gapdh*. (**B**) Relative integral value of the signal peak. The *db/db* mice value was set to 1.

**Table 1 cells-13-01419-t001:** Mapped data statistics in mRNA sequence.

Sample	Number of Mapped Reads (%)
db/m_1	47,501,746 (97.73)
db/m_2	43,732,867 (97.08)
db/m_3	48,231,655 (96.29)
db/m_4	44,039,336 (97.39)
db/db_1	47,501,746 (97.73)
db/db_2	41,823,389 (97.62)
db/db_3	44,158,962 (96.48)
db/db_4	41,520,847 (97.9)
db/db+RJ_1	39,459,436 (95.91)
db/db+RJ_2	41,564,058 (98.07)
db/db+RJ_3	46,891,376 (97.01)
db/db+RJ_4	41,309,857 (96.86)

**Table 2 cells-13-01419-t002:** Sequencing information of CUT&Tag.

Sample	Target	Total Number of Reads	Usable Number of Alignments	Used Number of Alignments	Number of Peaks
*db/m*	H3K27me3	34,840,978	13,892,705	11,279,709	70,350
*db/m*	H3K23Ac	28,296,100	12,356,384	11,279,699	67,071
*db/db*	H3K27me3	32,545,792	12,677,717	11,279,691	65,687
*db/db*	H3K23Ac	30,115,334	12,745,405	11,279,679	96,801
*db/db+RJ*	H3K27me3	33,225,536	11,279,720	11,279,669	51,294
*db/db+RJ*	H3K23Ac	33,789,784	14,173,301	11,279,660	98,254

## Data Availability

The datasets were uploaded at https://www.ncbi.nlm.nih.gov/bioproject/1143496 (accessed on 2 August 2024).

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
