# Peer review of "A Study of Small Intestinal Epigenomic Changes Induced by Royal Jelly"

_cells, 2024, doi:10.3390/cells13171419_

Round 1

Reviewer 1 Report

Comments and Suggestions for Authors

In this manuscript, Genki Kobayashi and colleagues explored the effects of royal jelly (RJ) on the small intestine. They first measured body weight, physical activity, insulin sensitivity, and lipid metabolism. They then utilized mRNA sequencing and CUT&Tag methods to explore histone modifications and gene expression changes. They found that RJ administration did not affect body weight but improved insulin sensitivity and lipid metabolism. RJ increased H3K27me3 and decreased H3K23Ac in genes associated with the G2M checkpoint, such as Smc2, Mcm3, Ccnd1, Rasal2, Mcm6, and Mad2l1.

Overall, I found this manuscript lacking organization, particularly in the experimental design and writing. The scientific writing is subpar, with the authors mostly listing the experiments performed without explaining their necessity. I detailed my comments as follows:

  1. The authors should provide the source of the db/m and db/db models. What are their strain numbers and genetic backgrounds? Why were these mouse models chosen to test the effects of RJ?
  2. Most figures are unclear, such as Fig 2A-2B and Fig 1C. The authors should provide high-resolution figures, especially for those with asterisk annotations.
  3. In line 198, regarding Fig 1C, the authors should state that the number of rotations was measured.
  4. Is there a mechanism linking metabolic disorders and the G2M checkpoint? The authors should review published research and provide a deeper analysis.
  5. The manuscript's writing should be improved. Firstly, the subtitles of the results section should summarize the key findings instead of merely listing the experiments. Secondly, the purpose of each experiment should be stated at the beginning of each paragraph, rather than abruptly transitioning from one experiment to another.
  6. Data availability for sequencing should be provided.
  7. Why was 5% RJ used for this study? Does RJ have a dose effect on the db/db mice?
  8. The analysis of the GO terms and KEGG pathways is too general. Lines 214-229 lack substantial information. Since the cell cycle was a significantly different signaling pathway, mentioning this pathway to link the G2M checkpoint and histone modifications would make more sense.
Comments on the Quality of English Language
  1. In line 198, regarding Fig 1C, the authors should state that the number of rotations was measured.

Reviewer 2 Report

Comments and Suggestions for Authors

This study provides insight into how royal Jelly (RJ) impacts the transcriptional state of intestinal cells. Similar studies have been done in the past showing similar results about the lowered expression of some genes and pathways in response to RJ consumption in mice. How RJ operates or causes these interesting transcriptional changes is not addressed in this manuscript. Overall, this is a very descriptive paper that provides a rationale for investigating this question further. It is missing mechanism and whether these transcriptional changes are organism-wide or just limited to intestinal cells (which is far less interesting therapeutically).

Major concerns:

Figure 1C the Y axis is broken at a place where it is impossible to determine the values for db/m light or db/db+RJ for the Dark cycle rotation. Please adjust to make these numbers more obvious in this graph.

Figure 2A is not legible. The provided text is invisible even at very high magnifications. This needs to be replaced. It is impossible to verity and evaluate these data.

The transcriptional changes that were observed were only limited to one tissue (Jejunum and/or colon) and then extrapolated for the whole organism. They either need to perform RNA-seq for at least 1-2 other tissues to show similar transcriptional changes or circumscribe their conclusions to this tissue type specifically throughout the paper. Even though suggestive, these data do NOT demonstrate that these transcriptional changes are happening to all tissues in the organisms.

Information about the depth of RNA-seq and ChIP-seq is missing. How many mapped reads per sample? How many biological replicates were used in the analyses?

The lowered expression of Smc2, Mcm3, Ccnd1, Rasal2, Mcm6, and Mad2l1 genes should be verified by qRT-PCR

There is no mention of genes whose expression increase in response to RJ. There are a few genes which display an increase, perhaps 3 (Figure 2C). The authors should provide some rationale for why RJ only leads to a decrease in gene expression over the entire transcriptome. Is there a component in RJ that is a known transcriptional repressor?

For Cut and tag-seq:

No rationale was provided about why the two histone modifications were selected ovef other histone marks (H3K27me and H3K23Ac). How many histone marks were tested? Also, information about the primary antibodies used for the ChIP assays is not provided. Are these antibodies the best commercially available ones for these modifications? What was n?

No error bars are shown in Figure 5B. How many biological replicates were performed? This can be used to calculate p values for these graphs.

Raw data for transcriptome and Cut and tag-seq are missing. These must be provided for independent evaluation and transparency.

Minor concerns.

Line 328 ‘….which was RELATED to histone modifications induced by RJ.’ Should be changed to ‘…which was CORRELATED WITH histone modifications induced by RJ.’ No cause an effect was shown; this is only a correlation.

Line 335 ‘…This finding suggests that RJ acts on small intestinal epithelial cells and causes EPIGENOMIC changes.’ Should be changed to ‘This finding suggests that RJ acts on small intestinal epithelial cells and causes CHANGES TO HISTONE MODIFICATIONS.’ There is no evidence that these changes are epigenetic and will persist after RJ is removed from their diet. The use of the term ‘epigenomic’ is incorrect and confusing.

Line 362-363 redundant clauses. Please correct this. There are several other instances of these mistakes; please correct.

Comments on the Quality of English Language

The paper needs extensive editing to remove text redundancies and correct spelling and typographic errors. Also, it would benefit from additional text describing the data and methods in more detail.

Round 2

Reviewer 1 Report

Comments and Suggestions for Authors

Thank you so much for the effort! The authors have addressed my concerns. 

Author Response

Thank you very much for your kind review.

Reviewer 2 Report

Comments and Suggestions for Authors

Similar to previous studies, this work reveals that in response to royal jelly (RJ) consumption, mouse intestinal cells lower the expression of some genes and pathways. How RJ operates or causes these interesting transcriptional changes is not addressed in this manuscript. However, this descriptive paper provides a rationale for investigating this question further. A key limitation of the study is lack of clearcut mechanism or basic understanding about how these transcriptional changes correlate with RJ consumption.

Major concerns:

Figure 2A is not legible again. In fact, the new version of this figure is even worse than the original version. Again, it is impossible to verity and evaluate these data. This is a major figure in the paper and HAS to be presented properly.

The lowered expression of Smc2, Mcm3, Ccnd1, Rasal2, Mcm6, and Mad2l1 genes should be verified by qRT-PCR.

This remains a serious concern. Because the cost associated with deep sequencing samples (already performed by the investigators) is far greater than ordering and testing a few primers, this argument is unacceptable. The qRT-PCR experiments can be done quickly and cheaply, providing key results that will further strengthen the conclusions of this paper.

There is no mention of genes whose expression increase in response to RJ. There are a few genes which display an increase, perhaps 3 (Figure 2C). The authors should provide some rationale for why RJ only leads to a decrease in gene expression over the entire transcriptome. Is there a component in RJ that is a known transcriptional repressor?

Thank you for your valuable comment. Sorry for the lack of explanation, the genes shown in

Figure 2C are not all the genes analyzed by mRNA-seq, but the genes set to be involved in the

G2M checkpoint by GSEA were extracted and shown in the volcano plots. Therefore, we did

not dare to mention the genes with increased expression this time.

The question remains though: what genes show an increase in their expression in intestinal cells of mice whose diet included RJ? What pathways are enriched? It is unclear why the authors do not provide these data or analyses. This remains an important component of this study and the data which should be analyses further. The authors also fail to provide access to raw data for independent verification of the results. Again, these remain serious concern, raising questions about lack of transparency.

For Cut and tag-seq:

No rationale was provided about why the two histone modifications were selected over other histone marks (H3K27me and H3K23Ac). How many histone marks were tested? Also, information about the primary antibodies used for the ChIP assays is not provided. Are these antibodies the best commercially available ones for these modifications? What was n?

Response

Thank you for your valuable comment. As you say, the reason why two histone modifications, H3K27me and H3K23Ac, were selected was missing. We analyzed histone modifications in small intestinal epithelial cells using Mod Spec®, a method that uses a mass spectrometer to analyze the relative amounts of more than 60 different histone modifications at once. The figure is newly added as Figure 3. Of these, those with significant differences between db/db and db/db+RJ mice, one from acetylation and one from methylation, and with primary antibodies for CUT&Tag were selected as H3K27me and H3K23Ac.

Once again, the data for Figure 3 remain illegible. Was H3K23Ac a mark tested in Figure 3? If so, it is not one of the modifications that shows a statistical change. This makes the connection to this histone modification illogical.

Materials and Methods

2.7. Mass Spectrometry for Histone Modifications (Mod-Spec)

“Small intestinal epithelial cells were isolated using previously published proto-cols [17].

Histone acetylation and methylation analysis were conducted using the Mod Spec method,

which includes histone extraction, protease digestion, and mass spec-trometry. This involved

a triple quadrupole mass spectrometer (TSQ Quantiva; Thermo Scientific) directly coupled

with a Dionex UltiMate 3000 nano-liquid chromatography system. The analysis was

performed by Active Motif (n = 4) [a].”

Results

3.6. Relative Abundance of Histone Modifications in Small Intestinal Epithelial Cells

“To determine whether RJ treatment causes changes in the chromatin structure of small

intestinal epithelial cells, changes in the relative amounts of posttranslational modifications

were examined between db/db and db/db+RJ mice by Mod Spec (Fig. 3). Histones

corresponding to antibodies that have been validated to perform well in the CUT&Tag assay

include H3K4me2, H3K4me3, H3K9ac, H3K27me3, and H3K36me3. Among them, H3K9ac

and H3K27me3 antibodies with statistically significant differ-ences were used for CUT&Tag.”

Overall, even though it is clear why the investigators sought to ascertain changes in H3K27me levels in their assay, the authors do not explain why they chose H3K23Ac for their analyses considering that their original assay (Figure 3) did not include or report this modification. This needs to be addressed much better and more clearly.

Figure legends

Figure 3. Histone modifications in small intestinal epithelial cells isolated from db/db and

db/db+RJ mice were determined by Mod Spec.

Heat map of the detected histone modifications. Histones in red have been validated to give good results in the CUT&Tag assay.

8. No error bars are shown in Figure 5B. How many biological replicates were performed?

This can be used to calculate p values for these graphs.

Response

Thank you for your valuable comment. As with RT-PCR, due to budget constraints, only one

sample each of H3K9ac and H3K27me3 antibodies could be performed on db/db and

db/db+RJ mice, respectively.

Once again, this response doesn’t make sense. Without error bars, this grant is not interpretable and creates a gap in confirmation of the key findings in the paper. These graphs in their current format (without error bars and n=1) are not suitable for publication.

9. Raw data for transcriptome and Cut and tag-seq are missing. These must be provided for

independent evaluation and transparency.

Response

Thank you for your valuable comment. We have already mentioned data availability statement

described as below.

Data Availability Statement: The datasets used during the current study are available from

the corresponding author on reasonable request.

The raw data should be made available to the reviewers of the paper for independent verification. Failing to do so raises questions about data transparency.

Comments on the Quality of English Language

N/A. The english is OK overall.

Author Response

Major concerns:   1. Figure 2A is not legible again. In fact, the new version of this figure is even worse than the original version. Again, it is impossible to verity and evaluate these data. This is a major figure in the paper and HAS to be presented properly.   Response Thank you for your valuable comment. According to your comment, we have modified the figure again.     2. The lowered expression of Smc2, Mcm3, Ccnd1, Rasal2, Mcm6, and Mad2l1 genes should be verified by qRT-PCR. This remains a serious concern. Because the cost associated with deep sequencing samples (already performed by the investigators) is far greater than ordering and testing a few primers, this argument is unacceptable. The qRT-PCR experiments can be done quickly and cheaply, providing key results that will further strengthen the conclusions of this paper.   Response Thank you for your valuable comment. According to your comment, we have performed the analyses of qRT-PCR described as below.   2. Materials and Methods 2.8. Quantitative Real Time-PCR The extraction of total RNA from the jejunum of murine models were converted to cDNA, and gene expression was analyzed with quantitative real time-PCR (qRT-PCR) (n = 4). The PCR primers used in this study, Smc2, Mcm3, Ccnd1, Rasal2, Mcm6, and Mad2l1I, are listed in Supplementary Table S1. The expression of each gene was normalized with Gapdh expression. The mRNA expression was measured with TaqMan Fast Advanced Master Mix (ThermoFisher Scientific, Waltham, MA, USA), according to the manufacturer’s protocol. The reactions were carried out in a StepOnePlus real-time PCR system (Applied Biosystems, Foster City, CA, USA) with the following thermal cycling conditions: 50°C for 2 minutes, 95°C for 20 seconds, followed by 40 cycles of 95°C for 1 second and 60°C for 20 seconds. The cycle threshold (CT) values were deter-mined using StepOnePlus Software v2.3 (Applied Biosystems). Data were analyzed us-ing the 2-ΔΔct method, and Smc2, Mcm3, Ccnd1, Rasal2, Mcm6, and Mad2l1I mRNA levels were calculated by comparing with db/db mice, each normalized to Gapdh mRNA as endogenous controls.   3. Results 3.5. The Expression of the Gene Set Associated with the G2M Checkpoint was Reduced by Roy-al Jelly Their gene expression analyzed by qRT-PCR in db/db+RJ mice was lower than that in db/db mice as well as mRNA-seq with statistical significance (Fig. 2D).     3. There is no mention of genes whose expression increase in response to RJ. There are a few genes which display an increase, perhaps 3 (Figure 2C). The authors should provide some rationale for why RJ only leads to a decrease in gene expression over the entire transcriptome. Is there a component in RJ that is a known transcriptional repressor? The question remains though: what genes show an increase in their expression in intestinal cells of mice whose diet included RJ? What pathways are enriched? It is unclear why the authors do not provide these data or analyses. This remains an important component of this study and the data which should be analyses further. The authors also fail to provide access to raw data for independent verification of the results. Again, these remain serious concern, raising questions about lack of transparency.   Response Thank you for your valuable comment. As you say, we have added the figures of gene set upregulated in db/db+RJ mice. In addition, we have uploaded the raw data of sequence.   3. Results 3.5. The Expression of the Gene Set Associated with the G2M Checkpoint was Reduced by Roy-al Jelly db/db mice were enriched for fatty acid metabolism, adipogenesis, bile acid metabo-lism, and other gene sets compared to db/m mice. enriched in db/db mice, whereas no enriched gene sets were detected in db/m mice (Supplementary Fig. S1A). Next, com-pared to db/db+RJ mice, gene sets related to the G2M checkpoint and coagulation were enriched in db/db mice (Supplementary Figure S1B). In addition, db/db+RJ mice had enriched gene sets associated with the p53 pathway and reactive oxigen species path-way (Supplementary Figure S1C). Since the gene sets involved in the G2M checkpoint was significantly enriched in db/db mice compared to db/m and db/db+RJ mice, we inves-tigated further (Figure 2B).   Data Availability Statement: The datasets were uploaded at https://www.ncbi.nlm.nih.gov/bioproject/1143496.   4. For Cut and tag-seq: No rationale was provided about why the two histone modifications were selected over other histone marks (H3K27me and H3K23Ac). How many histone marks were tested? Also, information about the primary antibodies used for the ChIP assays is not provided. Are these antibodies the best commercially available ones for these modifications? What was n? Once again, the data for Figure 3 remain illegible. Was H3K23Ac a mark tested in Figure 3? If so, it is not one of the modifications that shows a statistical change. This makes the connection to this histone modification illogical. Materials and Methods 2.7. Mass Spectrometry for Histone Modifications (Mod-Spec) “Small intestinal epithelial cells were isolated using previously published proto-cols [17]. Histone acetylation and methylation analysis were conducted using the Mod Spec method, which includes histone extraction, protease digestion, and mass spec-trometry. This involved a triple quadrupole mass spectrometer (TSQ Quantiva; Thermo Scientific) directly coupled with a Dionex UltiMate 3000 nano-liquid chromatography system. The analysis was performed by Active Motif (n = 4) [a].” Results 3.6. Relative Abundance of Histone Modifications in Small Intestinal Epithelial Cells “ To determine whether RJ treatment causes changes in the chromatin structure of small intestinal epithelial cells, changes in the relative amounts of posttranslational modifications were examined between db/db and db/db+RJ mice by Mod Spec (Fig. 3). Histones corresponding to antibodies that have been validated to perform well in the CUT&Tag assay include H3K4me2, H3K4me3, H3K9ac, H3K27me3, and H3K36me3. Among them, H3K9ac and H3K27me3 antibodies with statistically significant differ-ences were used for CUT&Tag.” Overall, even though it is clear why the investigators sought to ascertain changes in H3K27me levels in their assay, the authors do not explain why they chose H3K23Ac for their analyses considering that their original assay (Figure 3) did not include or report this modification. This needs to be addressed much better and more clearly.     Response Thank you for your valuable comment. According to your comment, first, we have made the heatmap large. Next, we have extracted histones that have been validated by the CUT&Tag assay and added a figure comparing their presence ratios between the two groups as Figure 3B. Because the differences between db/db and db/db+RJ mice were greatest for H3K23ac among acetylation and for H3K27me3 among methylation, further validation of those two histone modifications was performed.   3. Results 3.6. Relative Abundance of Histone Modifications in Small Intestinal Epithelial Cells Histones corresponding to antibodies that have been validated to perform well in the CUT&Tag assay include H3K4me2, H3K4me3, H3K9ac, H3K27me3, and H3K36me3. Because the differences between db/db and db/db+RJ mice were greatest for H3K23ac among acetylation and for H3K27me3 among methylation, further validation of those two histone modifications was performed (Fig. 3B).   Figure legends Figure 3. Histone modifications in small intestinal epithelial cells isolated from db/db and db/db+RJ mice were determined by Mod Spec. (B) Relative abundance of histone modification. The aver-age of the presene ratio of db/db mice was set to 1 (n = 4). Data are represented as the mean ± SD values. Data were analyzed using Welch’s t-tests. **p < 0.01 and ****p < 0.0001.      5. No error bars are shown in Figure 5B. How many biological replicates were performed? This can be used to calculate p values for these graphs. Response Thank you for your valuable comment. As with RT-PCR, due to budget constraints, only one sample each of H3K9ac and H3K27me3 antibodies could be performed on db/db and db/db+RJ mice, respectively. Once again, this response doesn’t make sense. Without error bars, this grant is not interpretable and creates a gap in confirmation of the key findings in the paper. These graphs in their current format (without error bars and n=1) are not suitable for publication.   Response Thank you for your valuable comment. In this study, RNA-seq analysis was performed at n=4 to identify differential gene expression. To complement this, a CUT&Tag assay was performed at n=1 to provide an initial assessment of specific epigenetic modifications; the CUT&Tag results are complementary and were used to initially confirm whether the changes in gene expression observed in RNA-seq were associated with epigenetic regulation. used to confirm whether the changes observed by RNA-seq are associated with epigenetic regulation. In the future, we plan to perform CUT&Tag analysis on more samples to validate these initial findings.   4. Discussion “”Furthermore, RNA-seq analysis was performed n=4 to identify important gene expres-sion changes, but the CUT&Tag assay performed to complement this was only n=1 per group. the CUT&Tag results are preliminary and intended to provide an initial as-sessment of epigenetic modifications. CUT&Tag results are preliminary and are in-tended to provide an initial assessment of epigenetic modifications. Therefore, the CUT&Tag data lack technical replicates and are insufficient for statistical validation. In future studies, the CUT&Tag assay should be performed on a larger number of sam-ples to improve the reproducibility and reliability of epigenetic changes. It is also im-portant to use other complementary methods (e.g., ChIP-Seq, ATAC-Seq) in combina-tion to analyze the observed epigenetic changes in more detail and to confirm the con-sistency of the results.”     6. Raw data for transcriptome and Cut and tag-seq are missing. These must be provided for independent evaluation and transparency. Response Thank you for your valuable comment. We have already mentioned data availability statement described as below. Data Availability Statement: The datasets used during the current study are available from the corresponding author on reasonable request. The raw data should be made available to the reviewers of the paper for independent verification. Failing to do so raises questions about data transparency.   Response Thank you for your valuable comment. We have uploaded the raw data of sequence.   Data Availability Statement: The datasets were uploaded at https://www.ncbi.nlm.nih.gov/bioproject/1143496.

Round 3

Reviewer 2 Report

Comments and Suggestions for Authors

Major concerns:

Figure 2A is not legible again. In the new PDF version of the paper, the figure, again, is not sufficiently of high resolution to show all the text. This has to be done correctly. This also applies to other figures in this paper.

Comments on the Quality of English Language

N/A

Author Response

Figure 2A is not legible again. In the new PDF version of the paper, the figure, again, is not sufficiently of high resolution to show all the text. This has to be done correctly. This also applies to other figures in this paper.

Response
Thank you for your kind comment. According to your comment, we have modified the figures.